# REDD1 Deletion Suppresses NF-κB Signaling in Cardiomyocytes and Prevents Deficits in Cardiac Function in Diabetic Mice

**DOI:** 10.3390/ijms25126461

**Published:** 2024-06-12

**Authors:** Shaunaci A. Stevens, Siddharth Sunilkumar, Sandeep M. Subrahmanian, Allyson L. Toro, Omer Cavus, Efosa V. Omorogbe, Elisa A. Bradley, Michael D. Dennis

**Affiliations:** 1Department of Cellular and Molecular Physiology, Penn State College of Medicine, Hershey, PA 17033, USA; 2Division of Cardiovascular Medicine, Penn State Health Heart and Vascular Institute, Hershey S. Milton Medical Center, Hershey, PA 17033, USA

**Keywords:** DDIT4, RTP801, inflammation, heart disease, diabetic cardiomyopathy

## Abstract

Activation of the transcription factor NF-κB in cardiomyocytes has been implicated in the development of cardiac function deficits caused by diabetes. NF-κB controls the expression of an array of pro-inflammatory cytokines and chemokines. We recently discovered that the stress response protein regulated in development and DNA damage response 1 (REDD1) was required for increased pro-inflammatory cytokine expression in the hearts of diabetic mice. The studies herein were designed to extend the prior report by investigating the role of REDD1 in NF-κB signaling in cardiomyocytes. REDD1 genetic deletion suppressed NF-κB signaling and nuclear localization of the transcription factor in human AC16 cardiomyocyte cultures exposed to TNFα or hyperglycemic conditions. A similar suppressive effect on NF-κB activation and pro-inflammatory cytokine expression was also seen in cardiomyocytes by knocking down the expression of GSK3β. NF-κB activity was restored in REDD1-deficient cardiomyocytes exposed to hyperglycemic conditions by expression of a constitutively active GSK3β variant. In the hearts of diabetic mice, REDD1 was required for reduced inhibitory phosphorylation of GSK3β at S9 and upregulation of IL-1β and CCL2. Diabetic REDD1^+/+^ mice developed systolic functional deficits evidenced by reduced ejection fraction. By contrast, REDD1^−/−^ mice did not exhibit a diabetes-induced deficit in ejection fraction and left ventricular chamber dilatation was reduced in diabetic REDD1^−/−^ mice, as compared to diabetic REDD1^+/+^ mice. Overall, the results support a role for REDD1 in promoting GSK3β-dependent NF-κB signaling in cardiomyocytes and in the development of cardiac function deficits in diabetic mice.

## 1. Introduction

Diabetes alters cardiac structure and function independent of coronary artery disease and hypertension, resulting in a metabolically triggered myopathic process known as diabetic cardiomyopathy [1]. Diabetes is characterized by chronic low-grade inflammation, which is a key factor responsible for the development of cardiovascular disease [2]. Myocardial activation of the transcription factor nuclear factor κB (NF-κB) is enhanced in heart failure patients [3]. The NF-κB family of transcription factors promotes the expression of genes involved in innate and adaptive immunity, cell survival, and inflammation (e.g., CCL2, IL-1β, and IL6) [4]. Generally, acute NF-κB activation, such as with preconditioning, is viewed as having a cardioprotective role, whereas chronic NF-κB signaling leads to heart failure [5]. While pre-clinical studies support that NF-κB activity is enhanced in both cardiac tissue from diabetic mice and in cardiomyocytes exposed to hyperglycemic conditions [6,7], the specific molecular mechanisms through which diabetes promotes NF-κB signaling in the heart have not been fully characterized.

Inactive NF-κB is sequestered in the cytoplasm by a family of proteins known as inhibitor of κB (IκB). In the canonical NF-κB signaling pathway, the IκB kinase complex (IKK) phosphorylates IκB to promote its proteasomal degradation and thus permit the IKK-dependent phosphorylation of NF-κB at S536 and NF-κB nuclear localization [4]. In support of a role for enhanced canonical NF-κB signaling in diabetes-induced heart disease, the expression of a phosphorylation-resistant IκB variant in the heart prevents cardiac function deficits in diabetic mice [8]. In addition to being activated by cytokine-mediated signaling, multiple signal transduction pathways influence IKK activity, allowing it to serve as a signaling integrator under a variety of physiological and pathophysiological conditions [3]. 

Our laboratory recently provided evidence that endoplasmic reticulum (ER) stress upregulates expression of the stress response protein regulated in DNA damage and development 1 (REDD1) in the hearts of diabetic mice and that REDD1 is required for increased pro-inflammatory cytokine expression in cardiomyocytes exposed to hyperglycemic conditions [9]. In the retina of diabetic mice, REDD1 acts to sustain canonical NF-κB signaling [10,11]. However, it is not known if REDD1 plays a similar role in the heart. REDD1 acts by targeting protein phosphatase 2A (PP2A) to Akt, leading to the site-specific dephosphorylation of Akt and reduced phosphorylation of its downstream targets [12]. Akt phosphorylates glycogen synthase kinase 3β (GSK3β) on its N-terminus at S9 to block substrate recognition [13]. In the retina of diabetic mice, REDD1 sustains IKK activity by promoting the activation of GSK3β [11]. Thus, REDD1 also potentially promotes NF-κB signaling in cardiomyocytes by enhancing GSK3β-dependent IKK activity. While this possibility has yet to be explored, GSK3β has been implicated in key cellular mechanisms that lead to several cardiomyopathies [14]. Indeed, a prior study supports a role for GSK3β in diabetes-induced cardiac inflammation and consequential remodeling [15]. The studies herein were designed to investigate the potential role of REDD1 in promoting GSK3β-dependent NF-κB signaling in cardiomyocytes and the consequent development of cardiomyopathy. 

## 2. Results

### 2.1. REDD1 Deletion Reduced NF-κB Signaling in Cardiomyocytes

To evaluate the role of REDD1 in NF-κB signaling, wild-type and REDD1-deficient AC16 cardiomyocytes were exposed to culture medium supplemented with TNFα. TNFα transiently enhanced phosphorylation of NF-κB p65 at S536 and IKKα/β at S179/S180 in both WT and REDD1-deficient cardiomyocytes (Figure 1A). Compared to WT cells, NF-κB p65 and IKKα/β phosphorylation in REDD1-deficient cells was attenuated after exposure to TNFα (Figure 1A–C). In WT cells exposed to TNFα for 4 h, NF-κB p65 was predominantly localized to the nucleus (Figure 1D). NF-κB p65 nuclear localization was less pronounced in REDD1-deficient cells, as compared to WT cells after TNFα exposure. Consistent with the reduction in NF-κB p65 nuclear localization, TNFα-induced expression of the mRNA encoding the pro-inflammatory chemokine *Ccl2* was also blunted in REDD1-deficient cells, as compared to WT cells (Figure 1E). The data support that REDD1 is necessary for sustained activation of NF-κB in cardiomyocytes. 

### 2.2. NF-κB Signaling Was Attenuated in REDD1-Deficient Cardiomyocytes Exposed to Hyperglycemic Conditions

Hyperglycemic conditions promote NF-κB signaling in AC16 cardiomyocytes [16]. To further investigate the impact of REDD1 on NF-κB activation, wild-type and REDD1-deficient AC16 cardiomyocytes were exposed to either hyperglycemic conditions or an osmotic control. Consistent with the suppressive effect of REDD1 deletion on TNFα-induced *Ccl2* mRNA expression, enhanced *Ccl2* mRNA expression in cells exposed to hyperglycemic conditions was also reduced by REDD1 deletion (Figure 2A). In AC16 cells exposed to hyperglycemic conditions, NF-κB p65 phosphorylation at S536 was attenuated in REDD1-deficient cells, as compared to wild-type cells (Figure 2B,C). Hyperglycemic conditions promoted dramatic nuclear localization of NF-κB p65 in wild-type cells, and the effect was reduced in REDD1-deficient cells (Figure 2D). The data support a critical role for REDD1 in cardiomyocyte NF-κB activation in response to hyperglycemic conditions. 

### 2.3. GSK3β Promoted NF-κB Signaling in Cardiomyocytes Exposed to Hyperglycemic Conditions

To evaluate the role of GSK3β in REDD1-dependent cardiomyocyte NF-κB activation, GSK3β expression was suppressed by shRNA knockdown. Consistent with the effect of REDD1 deletion, GSK3β knockdown suppressed *Ccl2* mRNA expression in AC16 cardiomyocytes exposed to culture medium supplemented with TNFα (Figure 3A). GSK3β knockdown also blunted phosphorylation of NF-κB p65 and IKKα/β in cells exposed to hyperglycemic conditions (Figure 3B,C). *Ccl2*, *Il1b*, and *Il6* mRNA expression were reduced by GSK3β knockdown in cardiomyocyte cultures exposed to hyperglycemic conditions (Figure 3D). To evaluate the role of GSK3β in the suppressive effect of REDD1 deletion on NF-κB signaling, a constitutively active GSK3β variant that resists Akt-dependent inhibitory phosphorylation was expressed in REDD1-deficient cells. NF-κB reporter activity was blunted in REDD1-deficient cells exposed to hyperglycemic conditions, as compared to wild-type cells (Figure 3E). Importantly, expression of the constitutively active GSK3β variant was sufficient to restore NF-κB activity in REDD1-deficient cells exposed to hyperglycemic conditions. Together, the data support a role for GSK3β in REDD1-dependent NF-κB activation.

### 2.4. REDD1 Was Necessary for Reduced Phosphorylation of GSK3β in the Hearts of Diabetic Mice

In the hearts of diabetic REDD1^+/+^ mice, the inhibitory phosphorylation of GSK3β at S9 was reduced, as compared to non-diabetic mice (Figure 4A). Unlike REDD1^+/+^ mice, diabetic REDD1^−/−^ did not exhibit reduced phosphorylation of GSK3β at S9 (Figure 4B). Consistent with the change in GSK3β phosphorylation, CCL2 and IL1β were also upregulated in the hearts of diabetic mice, and REDD1 was required for the diabetes-induced increase (Figure 4C–E). The data support that REDD1 is necessary for GSK3β activation and increased pro-inflammatory response in the hearts of diabetic mice. 

### 2.5. REDD1 Contributed to Cardiac Defects in Diabetic Mice

Diabetic REDD1^+/+^ and REDD1^−/−^ mice exhibited similar increases in fasted blood glucose concentrations (Figure 5A) and a modest reduction in body weight (Figure 5B). Diabetes promoted REDD1 mRNA expression in the hearts of REDD1^+/+^ mice, whereas REDD1 was absent from the hearts of REDD1^−/−^ mice (Figure 5C). In coordination with the increase in REDD1, diabetes enhanced *Ccl2* mRNA expression in the hearts of REDD1^+/+^ mice but not in REDD1^−/−^ mice (Figure 5D). To determine if the REDD1-dependent signaling changes in the heart of diabetic mice were associated with a functional deficit, echocardiography was performed (Figure 5E). The heart rate was not different between groups (Figure 5F). Compared to diabetic REDD1^−/−^ mice, diabetic REDD1^+/+^ mice demonstrated ventricular remodeling, hallmarked by chamber dilation (LVID, d 0.45 ± 0.05 vs. 0.38 ± 0.02 mm, *p* = 0.016; Figure 5G,H) and evidence of systolic dysfunction (Figure 5I,J). In particular, left ventricular ejection fraction was reduced in diabetic REDD1^+/+^ mice, as compared to non-diabetic REDD1^+/+^ mice (LVEF 41 ± 10 vs. 50 ± 3, *p* = 0.015). On the contrary, ejection fraction and fractional shortening were similar in diabetic REDD1^−/−^ mice, as compared to non-diabetic REDD1^−/−^ mice. Moreover, there was no significant difference in left ventricular remodeling or systolic function in non-diabetic REDD1^+/+^ and REDD1^−/−^ mice. Together, the data support that REDD1 is necessary for developing early dilated cardiomyopathy in diabetic mice. 

## 3. Discussion

Heart failure is the leading cause of mortality and morbidity in people living with diabetes [17]. Prior studies support that myocardial NF-κB signaling plays a central role in the development of the functional deficits that are caused by diabetes [8,18]. Herein, REDD1 deletion protected diabetic mice from cardiac dysfunction. These proof-of-concept studies implicate a REDD1/GSK3β/NF-κB signaling axis in the pro-inflammatory response of cardiomyocytes to diabetes. The data are consistent with a growing body of work that establishes a key role for REDD1 in the activation of inflammatory pathways by promoting NF-κB signaling [11,19,20,21]. Overall, the studies are consistent with a working model in which REDD1-dependent NF-κB activation in cardiomyocytes promotes diabetes-induced cardiac dysfunction (Figure 6).

NF-κB activation is increased in the myocardium of heart failure patients in coordination with upregulated expression of NF-κB gene targets [3]. Similarly, NF-κB nuclear localization is increased in the hearts of diabetic mice [8], and evidence supports increased NF-κB DNA-binding in the hearts of diabetic rats [6,22]. NF-κB stimulates the expression of pro-inflammatory factors, including cytokines, acute-phase proteins, and chemokines, which contribute to cardiovascular disease by promoting inflammation, endothelial dysfunction, and atherosclerosis [18]. NF-κB also enhances the production of extracellular matrix proteins, leading to cardiac fibrosis and scar formation [18]. NF-κB signaling is increased in cardiomyocytes exposed to hyperglycemic conditions, resulting in structure and function changes [7]. In cardiomyocyte cultures exposed to hyperglycemic conditions, REDD1 protein abundance was increased in coordination with an increase in NF-κB signaling. Importantly, REDD1 deletion reduced NF-κB signaling, nuclear localization of the transcription factor, and cytokine expression levels in cardiomyocytes exposed to hyperglycemic conditions. 

A role for REDD1-dependent NF-κB signaling in cardiomyocytes supports prior works demonstrating a role for REDD1 in diabetes-induced pro-inflammatory signaling. Specifically, REDD1 is necessary for increased NF-κB activation and enhanced pro-inflammatory cytokine expression in the retina of diabetic mice [10]. REDD1 deletion also prevents NF-κB activation in adipocytes and reduces inflammatory cytokine levels in plasma of diabetic mice [20]. Together, the studies highlight a central role for REDD1 in the development of the chronic low-grade inflammatory state that characterizes diabetes. 

ER stress and inflammation are two well-established markers of heart disease [23]. In the hearts of diet-induced obese rats [24] and in mice fed a pro-diabetogenic diet [9], REDD1 expression is increased in coordination with markers of ER stress and inflammation. ER stress activates the unfolded protein response (UPR) to restore ER homeostasis. This is achieved in part by activation of the kinase protein kinase R-like endoplasmic reticulum kinase (PERK), which phosphorylates the α-subunit of the translation initiation factor eIF2, leading to the preferential translation of activating transcription factor 4 (ATF4) [25]. ATF4 promotes the expression of several stress response genes, including REDD1 [26]. In cardiomyocytes exposed to hyperglycemic conditions, PERK inhibition or ATF4 knockdown prevents increased REDD1 expression [9]. Moreover, REDD1 deletion prevents an increase in pro-inflammatory cytokine expression in cardiomyocytes exposed to hyperglycemic conditions and in the hearts of diabetic mice [9]. Together, the data support that REDD1 acts as an important molecular link between myocardial ER stress and inflammation in the context of diabetes. 

Lee et al. found that REDD1 promotes atypical NF-κB signaling by directly interacting with IκB and preventing the formation of the inhibitory NF-κB·IκB complex [20,27]. Consequently, increased REDD1 expression is potentially sufficient to promote NF-κB nuclear localization independently of stimuli that activate cell surface receptors or the classic NF-κB signaling pathways. In cardiomyocytes exposed to hyperglycemic conditions, REDD1 deletion reduced IKK autophosphorylation. REDD1-dependent IKK autophosphorylation is also seen in the retina of diabetic mice and retinal cells exposed to hyperglycemic conditions [10,28]. Importantly, REDD1-dependent IκB sequestration does not explain the effect of REDD1 on IKK autophosphorylation in response to diabetes. Alternatively, the studies herein are consistent with our prior work demonstrating that REDD1 also acts to enhance NF-κB signaling by promoting GSK3β-dependent IKK complex assembly and activation of the kinase [10,11]. 

GSK3 signaling has been implicated as having both protective and detrimental roles in myocardial disease [29]. Diabetes enhances the activation of GSK3β in the heart, as evidenced by reduced inhibitory phosphorylation at S9 [30]. We observed a similar decrease in GSK3β phosphorylation in the hearts of diabetic mice, which was dependent on REDD1. Interventions that restore GSK3β phosphorylation in the heart of diabetic rats are associated with a reduction in the apoptosis and hypertrophy induced by diabetes [30]. In fact, GSK3β pharmacologic inhibition prevents diabetes-induced changes in inflammatory markers and cardiac remodeling [15]. Hyperglycemic conditions also decrease GSK3β phosphorylation in cardiomyocyte cultures [31,32]. In the studies here, GSK3β knockdown reduced NF-κB signaling and pro-inflammatory cytokine expression in cardiomyocytes exposed to hyperglycemic conditions. Moreover, expression of a constitutively active GSK3β variant restored NF-κB activity in REDD1-deficient cardiomyocytes exposed to hyperglycemic conditions. The observation suggests that the protective effects of REDD1 deletion are achieved via GSK3β downregulation. It is important to note that cardiomyocyte-specific GSK3β deletion exacerbates cardiac dysfunction in obese mice [33]. Thus, properly regulated GSK3 signaling is likely critical for balancing adaptive versus maladaptive processes in the heart. The studies here support a potential role for REDD1-dependent GSK3 signaling in increased pro-inflammatory cytokine expression and cardiac functional deficits caused by diabetes.

We found that REDD1 deletion was sufficient to protect diabetic mice from developing early dilated cardiomyopathy, evidenced by protection from chamber dilation and reduction in ejection fraction. A prior report demonstrated that a 5-fold increase in cardiac REDD1 expression via AAV infection increases both left ventricular end-diastolic diameter (LVEDd) and left ventricular end-systolic diameter (LVESd), indicating that REDD1 may mediate the development of eccentric hypertrophy [34]. Moreover, partial REDD1 knockdown (<50%) prevents an increase in LVEDd and LVESd in a murine model of doxorubicin-induced cardiotoxicity, potentially indicating it is also important in mediating other causes of non-ischemic dilated cardiomyopathy [34]. Notably, in cardiomyocytes exposed to doxorubicin, REDD1 knockdown reduces NF-κB nuclear localization [34]. The studies herein extend on the prior report by providing evidence for REDD1-dependent proinflammatory signaling in the progression of diabetes-induced cardiac functional changes. 

Overall, the studies here support that REDD1 is necessary for the development of eccentric hypertrophy in diabetic mice. The data suggest that inhibition of the REDD1/GSK3β/NF-κB signaling axis in cardiomyocytes could be a potential therapeutic strategy to prevent dilated cardiomyopathy. This could be achieved by either REDD1 silencing or inhibition of GSK3β/NF-κB activity. One limitation of the proposed studies is that REDD1 deletion in mice was not cardiomyocyte-specific. Thus, the absence of REDD1 specifically in cardiomyocytes versus immune cells of the heart or other systemic factors in preventing cardiac function deficits in diabetic mice remains to be established. Additionally, while diabetic mice and cell cultures exposed to hyperglycemic conditions are helpful experimental models to identify key signaling mechanisms, they do not fully recapitulate the complex disease pathology that leads to functional decline in heart disease patients. Regardless, the data herein warrant further investigation of REDD1/GSK3β/NF-κB signaling in diabetic cardiomyopathy.

## 4. Materials and Methods

### 4.1. Cell Culture

Human AC16 adult ventricular cardiomyocytes were purchased from Sigma-Aldrich (St. Louis, MO, USA, SCC109, RRID: CVCL_4U18). REDD1-deficient AC16 cells were generated by CRISPR/Cas9 using a pLentiCRISPR v.2 construct containing a REDD1 guide RNA, as described previously [9]. AC16 cells were maintained in Dulbecco’s modified Eagle’s medium/Nutrient Mixture F-12 (Gibco, Grand Island, NY, USA, 11320082) supplemented with 10% fetal bovine serum (FBS, Atlas Biologicals) and 1% penicillin/streptomycin (P/S) in a humidified incubator with 5% CO_2_ at 37 °C. Where indicated, cell culture medium was supplemented with 25 ng/mL human recombinant TNFα (Sigma). F-12 medium contains 17 mM glucose. Thus, to evaluate the impact of hyperglycemic conditions, cells were initially adapted to Dulbecco’s modified Eagle’s medium (DMEM, Gibco 11885092) containing 5 mM glucose for at least 2 passages, as previously described [9]. Cells were then exposed to culture medium supplemented with either 25 mM glucose or 5 mM glucose and 20 mM mannitol as an osmotic control for up to 24 h. Cell transfections were performed with jetPRIME (Polyplus). Plasmids included pCMV5 vector, pCMV-HA-caGSK3β, NF-κB-TATA luciferase reporter plasmid, and pRL-Renilla luciferase (Promega, Madison, WI, USA). GSK3β knockdown was achieved with pLKO-shGSK3β plasmids obtained from the Penn State College of Medicine TRC1 Human Library Informatics Core (shRNA TRC ID# TRCN0000000822). Lentivirus containing an shRNA targeting GSK3β (sequence: 5′-CCGGCCCAAATGTCAAACTACCAAACTCGAGTTTGGTAGTTTGACATTTGGGTTTTT-3′) was prepared from HEK293FT cells and used to infect AC16. Cells with stable shRNA expression were selected with puromycin (2 µg/mL). The Dual-Luciferase Assay Kit (Promega) was used to measure luciferase activity in cells expressing NF-κB-TATA luciferase and pRL-Renilla luciferase plasmids on a FlexStation3 (Molecular Devices, San Jose, CA, USA). 

### 4.2. Protein Analysis

AC16 cells were collected in sodium dodecyl sulfate (SDS) sample buffer, boiled for 5 min, and analyzed by Western blotting. Proteins were fractionated in Criterion Precast 4–20% gels (Bio-Rad Laboratories, Hercules, CA, USA) and transferred to polyvinylidene fluoride (PVDF). The PVDF membrane (Bio-Rad Laboratories) was blocked with 5% milk in Tris-buffered saline Tween 20 (TBS-T), washed, and incubated overnight at 4 °C with the appropriate primary antibody (Appendix A). The PVDF membrane was then washed in TBS-T and then exposed to the appropriate secondary antibody in 5% milk TBS-T. The blot was washed in TBS-T, and the antigen–antibody interaction was visualized with enhanced chemiluminescence Clarity Reagent (Bio-Rad Laboratories) using a FluorChem E imaging system (ProteinSimple, San Jose, CA, USA). 

### 4.3. PCR Analysis

RNA was extracted from cells or cardiac tissue using TRIzol (Invitrogen, Waltham, MA, USA) according to the manufacturer’s protocol. RNA (1 μg) was reverse transcribed using a High-Capacity cDNA Reverse Transcription Kit (Applied Biosystems, Waltham, MA, USA) and subjected to quantitative real-time PCR (qRT-PCR) (QuantStudio 12K Flex Real-Time PCR System, RRID:SCR_021098) with QuantiTect SYBR Green master mix (Qiagen, Hilden, Germany). Primer sequences are listed in Appendix A. Mean cycle threshold (*C_T_*) values were determined for each sample. Changes in mRNA expression were normalized to actin mRNA using the 2^−ΔΔ*CT*^ calculations. 

### 4.4. Animals

Diabetes was induced in 6-week-old B6;129 REDD1^+/+^ and REDD1^−/−^ male mice [35] by intraperitoneal injection of 50 mg/kg streptozotocin (STZ) for 5 consecutive days, as previously described [36]. Non-diabetic mice were administered a sodium citrate buffer control. In some studies, diabetic mice were also fed a pro-diabetogenic high-fat high-sucrose (HFHS) diet (TD.88137) containing 42% kcal from fat, 42% kcal from carbohydrates, and 15.2% from protein (Envigo, Huntingdon, UK) to exacerbate diabetes phenotype. Control mice were fed a Teklad control diet (TD.08485). Diets were initiated at 6 weeks of age, and 2 weeks later, STZ was administered, as described above. Diabetic phenotype was determined by fasting blood glucose concentration >250 mg/dL 2 weeks after STZ administration. After 16 weeks of intervention, mice were euthanized, and cardiac tissue was collected for analysis. 

### 4.5. Immunofluorescent Microscopy

Hearts were excised, washed with ice-cold PBS, and immediately incubated for 24 h in 4% paraformaldehyde (PFA, pH 7.5). Hearts were then cut along the longitudinal axis, dehydrated in ethanol-xylene, and embedded in paraffin. Sections (6 µm) were cut from formalin-fixed, paraffin-embedded (FFPE) hearts. Following hydration to water, heat-induced antigen retrieval was carried out with citrate buffer (0.01 M, pH 6), and endogenous peroxidase activity was quenched by incubating sections in BLOXALL (Vector labs, Newark, CA, USA) blocking solution. Sections were blocked with 1% horse serum, followed by overnight incubation with the antibodies listed in Appendix A. Sections were incubated with appropriate secondary antibodies and counter-stained with 1 μmol/L DAPI. Slides were mounted with Fluoromount aqueous mounting media (Sigma-Aldrich, St. Louis, MO, USA) and imaged with a Leica SP8 confocal laser microscope (Leica, Wetzlar, Germany) using frame-stack sequential scanning. 

### 4.6. Echocardiography

Mice were subjected to noninvasive transthoracic 2-D echocardiography using a GE Logiq e (GE Healthcare, Wauwatosa, WI, USA) small animal echocardiography machine with the L10-22 (mHz) transducer, as described before [37]. Initially, mice were anesthetized with 2.5% isoflurane in 95% O_2_ and 5% CO_2_ and maintained with 1.25% isoflurane during data acquisition. Two-dimensional parasternal short-axis images were used to generate an M-mode view at the level of the papillary muscles, and end-diastolic frames were used to measure septal thickness, left ventricular internal diameter in both systole and diastole (LVID, d; LVID, s), and posterior and anterior wall thickness (LVPW; LVAW). Ejection fraction (EF%) and fractional shortening (FS%) were assessed from the parasternal long-axis and parasternal short-axis M-mode measurements.

### 4.7. Data Analysis and Presentation

GraphPad Prism software (version 9.3.0 (345)) was used to perform statistical analyses. Data were analyzed by either Student’s *t*-test or two-way analysis of variance. Trend tests and pairwise comparisons were conducted with the Tukey test for multiple comparisons. Graphical abstract and working model graphics were created using BioRender.com (accessed on 1 May 2024).

## Figures and Tables

**Figure 1 ijms-25-06461-f001:**
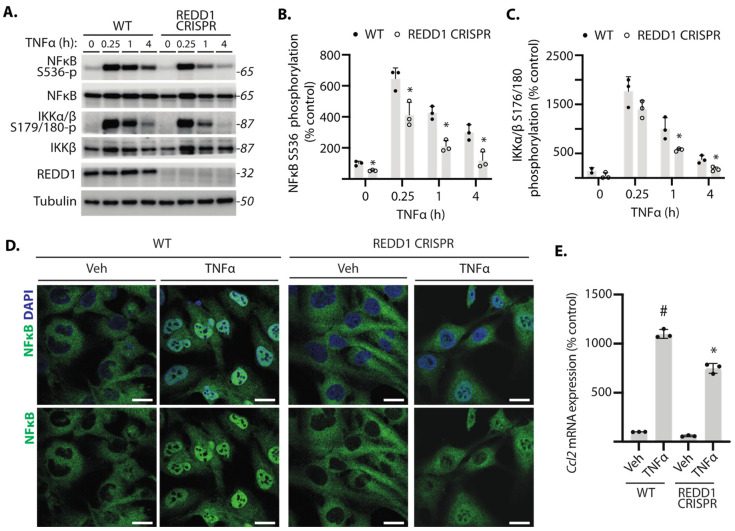
**REDD1 deletion attenuated NF-κB signaling in cardiomyocytes.** Human wild-type (WT) and REDD1 CRISPR AC16 cardiomyocytes were exposed to culture medium supplemented with the cytokine TNFα. (**A**–**C**), Western blotting was used to evaluate the phosphorylation of NF-κB p65 at S536 and IKKα/β at S179/180. Representative blots are shown with protein molecular mass in kDa at *right*. Phosphorylation of NF-κB p65 and IKK in (**A**) are quantified in (**B**,**C**), respectively. (**D**), Cardiomyocytes were exposed to TNFα or vehicle (Veh) control for 4 h, and NF-κB p65 (*green*) nuclear localization was evaluated by immunofluorescence. Nuclei were labeled with DAPI (*blue*). Representative micrographs are shown (scale bar: 25 µm). (**E**), *Ccl2* mRNA expression was determined in cell lysates by RT-PCR after 4 h of exposure to TNFα or Veh. Data are presented as means ± SD with individual data points plotted. Differences between groups were identified by two-way ANOVA. *, *p* < 0.05 versus WT; # *p* < 0.05 versus Veh.

**Figure 2 ijms-25-06461-f002:**
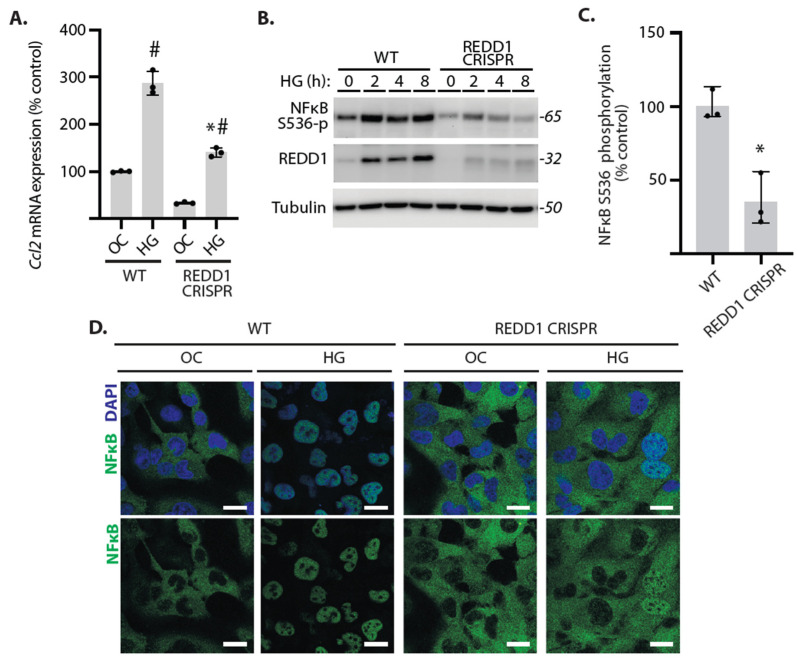
**NF-κB nuclear localization in response to hyperglycemic conditions was reduced in REDD1-deficient cardiomyocytes.** Human wild-type (WT) and REDD1 CRISPR AC16 cardiomyocytes were exposed to culture medium containing either 25 mM glucose (HG) or 5 mM glucose with 20 mM mannitol as an osmotic control (OC). (**A**), Cells were exposed to HG or OC for 24 h and RT-PCR was used to quantify *Ccl2* mRNA expression. (**B**,**C**), Phosphorylation of NF-κB p65 at S536 was evaluated by Western blotting. Representative blots are shown with protein molecular mass in kDa shown at *right*. NF-κB p65 phosphorylation in (**B**) is quantified in (**C**). (**D**), NF-κB p65 (*green*) nuclear localization was evaluated by immunofluorescence after 16 h of exposure to HG or OC. Nuclei were visualized with DAPI (*blue*). Representative micrographs are shown (scale bar: 25 µm). Data are presented as means ± SD with individual data points plotted. Differences between groups were identified by two-way ANOVA or Student’s *t*-test. *, *p* < 0.05 versus WT; # *p* < 0.05 versus OC.

**Figure 3 ijms-25-06461-f003:**
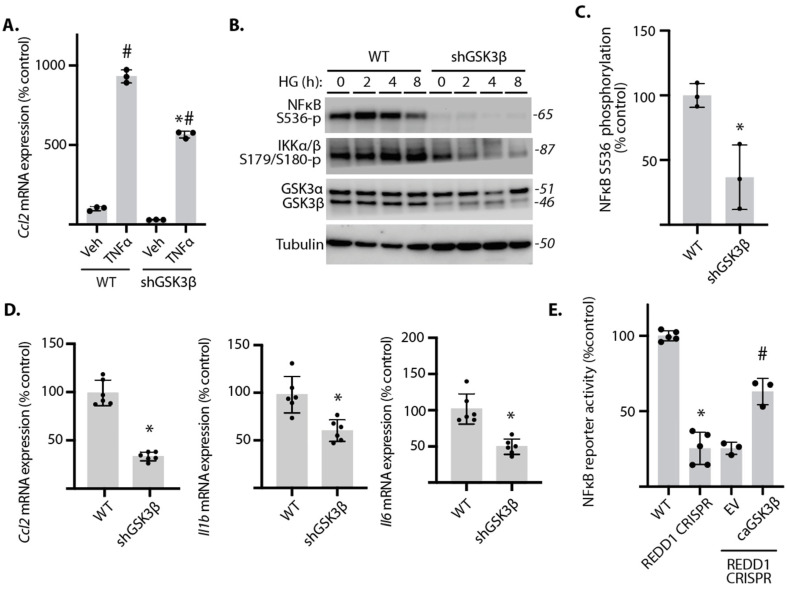
**GSK3β was required for enhanced NF-κB signaling in cardiomyocytes exposed to hyperglycemic conditions**. GSK3β was knocked down in human AC16 cardiomyocytes by stable expression of a shRNA (shGSK3β). (**A**), Wild-type (WT) and shGSK3β AC16 cells were exposed to culture medium supplemented with either TNFα or vehicle (Veh) for 4 h and *Ccl2* mRNA expression was determined by RT-PCR. (**B**,**C**), Cells were exposed to culture medium containing either 25 mM glucose (HG) or 5 mM glucose with 20 mM mannitol as an osmotic control (OC). Phosphorylation of NF-κB p65 at S536 and IKKα/β at S179/180 was evaluated by Western blotting. Representative blots are shown with protein molecular mass in kDa shown at *right*. NF-κB p65 phosphorylation after 8 h of HG in (**B**) is quantified in (**C**). (**D**), Expression of mRNAs encoding *Ccl2*, *Il1b*, and *Il6* was determined in cells exposed to HG for 24 h. (**E**), NF-κB activity was measured in lysates from AC16 cells expressing NF-κB firefly luciferase/*Renilla* luciferase reporter plasmids by dual luciferase assay after 16 h exposure to HG. Data are presented as means ± SD with individual data points plotted. Differences between groups were identified by two-way ANOVA or Student’s *t*-test. *, *p* < 0.05 versus WT; # *p* < 0.05 versus Veh or EV.

**Figure 4 ijms-25-06461-f004:**
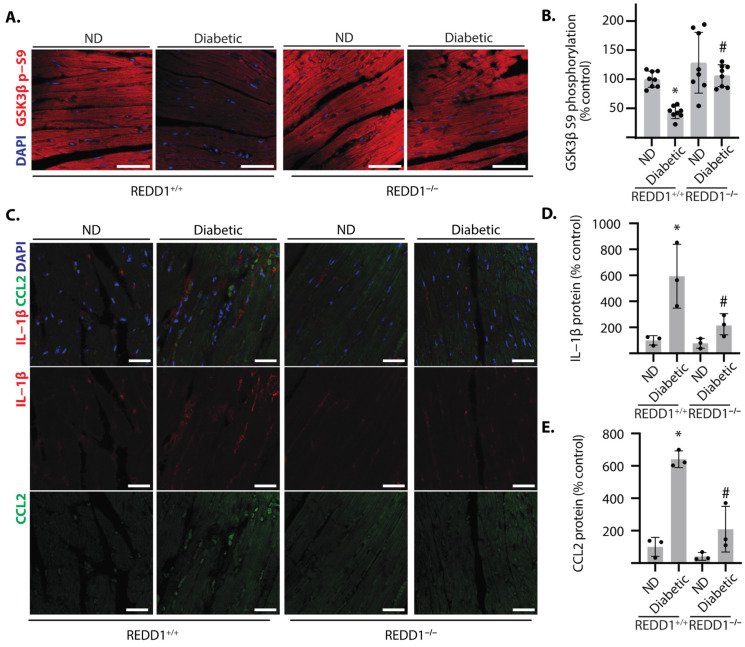
**REDD1 promoted GSK3β dephosphorylation and cytokine expression in the hearts of diabetic mice.** Diabetes was induced in REDD1^+/+^ and REDD1^−/−^ mice by administration of streptozotocin. Non-diabetic (ND) mice were administered vehicle injections. Cardiac sections were prepared after 16 weeks of diabetes. (**A**), GSK3β phosphorylation at S9 (*red*) was determined by immunolabeling of cardiac sections. Nuclei were visualized with DAPI (*blue*). Representative micrographs are shown (scale bar: 50 µm). (**B**), GSK3β S9 phosphorylation in (**A**) was quantified. (**C**), IL-1β (*red*) and CCL2 (*green*) were visualized in cardiac sections. (**D**,**E**), IL-1β (**D**) and CCL2 (**E**) protein in (**C**) were quantified. Individual data points are plotted with values presented as means ± SD. Differences between groups were identified by two-way ANOVA. *, *p* < 0.05 versus ND; # *p* < 0.05 versus REDD1^−/−^.

**Figure 5 ijms-25-06461-f005:**
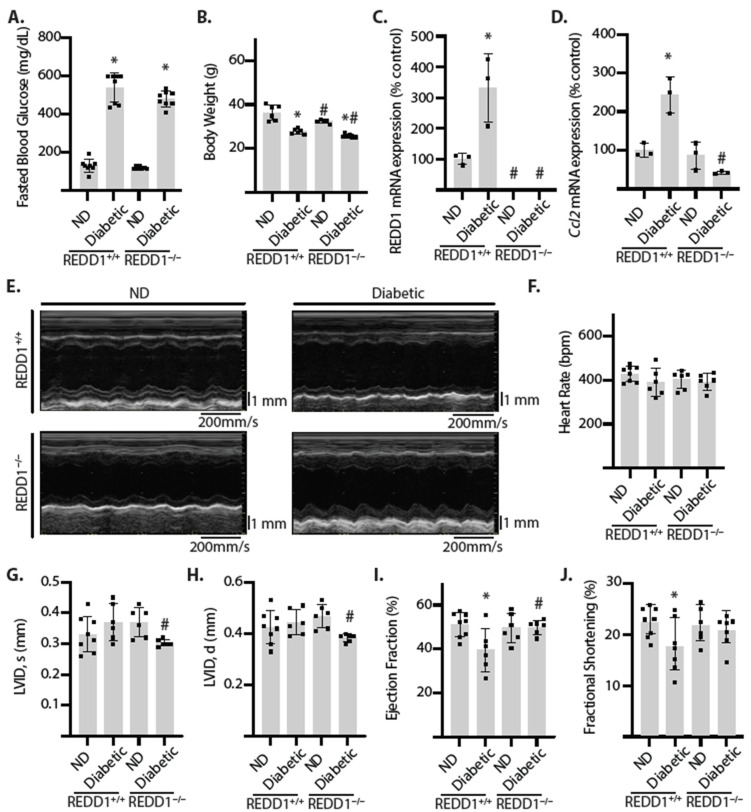
**Cardiac function was preserved in diabetic REDD1-deficient mice**. Diabetes was induced in REDD1^+/+^ and REDD1^−/−^ mice by streptozotocin administration and feeding a pro-diabetogenic diet. Non-diabetic (ND) mice were administered vehicle injections and fed a standard chow diet. Mice were evaluated after 16 weeks of intervention. (**A**), Blood glucose concentrations were measured following 4 h fast. (**B**), Body weights were determined. (**C**,**D**), REDD1 (**C**) and *Ccl2* (**D**) mRNA expression were determined in cardiac tissue homogenates by RT-PCR. (**E**–**J**), Echocardiography was performed. Representative left ventricular M-mode tracings are shown. Heart rate (**F**) left ventricular internal diameter at systole-end (LVID, s; (**G**), left ventricular internal diameter at diastole-end (LVID, d; (**H**), ejection fraction (**I**), and fractional shortening (**J**) were calculated. Individual data points are plotted with values presented as means ± SD (n = 3–8 mice per group). Differences between groups were identified by two-way ANOVA. *, *p* < 0.05 versus ND; # *p* < 0.05 versus REDD1^+/+^.

**Figure 6 ijms-25-06461-f006:**
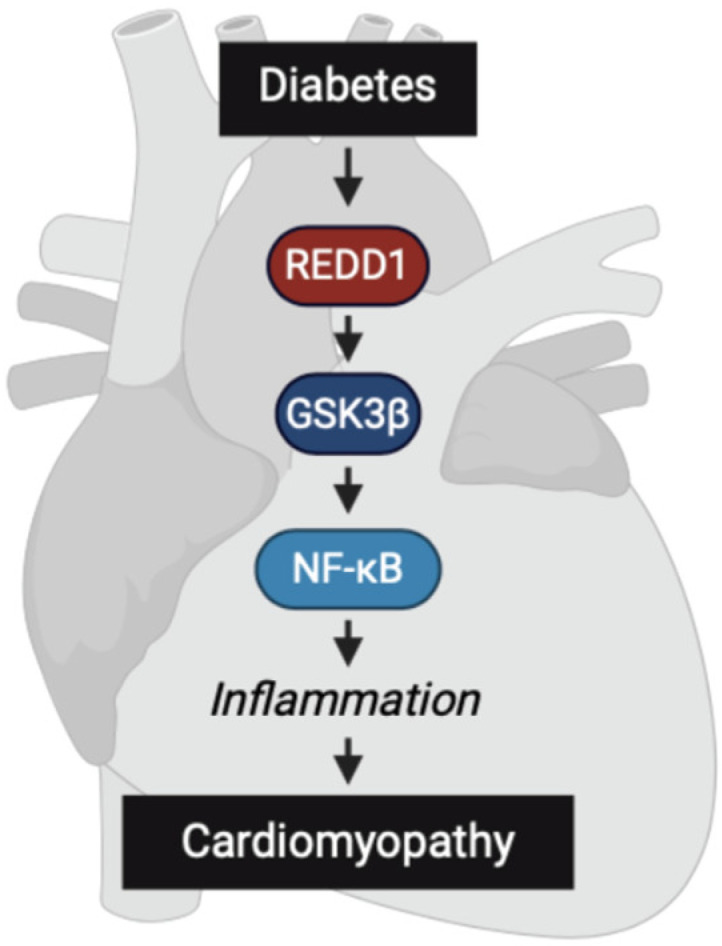
**Working model for the role of a REDD1/GSK3β/NF-κB signaling axis in diabetes-induced cardiac function deficits.** REDD1 was necessary for enhanced GSK3β signaling, NF-κB activation, and pro-inflammatory cytokine expression in cardiomyocytes exposed to hyperglycemic conditions and in the development of cardiac function deficits in diabetic mice.

## Data Availability

Primary data supporting the findings of this study are openly available at https://doi.org/10.6084/m9.figshare.25832830.v1 (accessed on 17 May 2024).

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
