# Peer review of "REDD1 Deletion Suppresses NF-κB Signaling in Cardiomyocytes and Prevents Deficits in Cardiac Function in Diabetic Mice"

_ijms, 2024, doi:10.3390/ijms25126461_

Round 1
Reviewer 1 Report
Comments and Suggestions for Authors
Authors studied the potential role of redd1 in diabetic cardiomyopathy, particularly on the inflammatory response of cardiomyocytes. The work suggests an interesting pathway for future treatment, but some issues must be corrected or added.
- The nfkb subunit which is phosphorylated nfkb should be specified. P65 can be the active subunit to bind to promoter regions for gene regulation
- In general, phosphorylated levels of mediators should be compared with those non-phosphorylated proteins
- The most intriguing issue can be the activation of redd1 after hyperglycemic insult. Could you discuss potential receptors and mediators (i.e., transcription factors)?
- How would cells react after HG and both gsk3b/redd1 silencing?
Methods:
- Why did you use different tools for silencing for gsk3b and redd1?
- F12 media has 17mM glucose, which can be considered relatively high. Why did not you grow them at F12-5 mM rather than changing the media to DMEM for 2 passages?
- Did animals change the body weight after stz+/- redd1 ko?
Author Response
Reviewer 1: Authors studied the potential role of redd1 in diabetic cardiomyopathy, particularly on the inflammatory response of cardiomyocytes. The work suggests an interesting pathway for future treatment, but some issues must be corrected or added.
- The nfkb subunit which is phosphorylated nfkb should be specified. P65 can be the active subunit to bind to promoter regions for gene regulation.
Response: The manuscript text and Figure Legends 1-3 have been edited to more clearly reflect that the p65 subunit of NF-κB is specifically evaluated in each of the experimental assessments.
- In general, phosphorylated levels of mediators should be compared with those non-phosphorylated proteins.
Response: As suggested, western blots for non-phosphorylated NF-κB p65 and IKKβ protein were added to the analysis in Figs. 1A. We did not find that total protein levels changed in the experimental conditions tested here.
- The most intriguing issue can be the activation of redd1 after hyperglycemic insult. Could you discuss potential receptors and mediators (i.e., transcription factors)?
Response: As suggested, we have revised the Discussion to more clearly outline the signaling events whereby hyperglycemic conditions promote REDD1 expression. Please see page 9 lines 239-252 of the revised manuscript. Specifically, hyperglycemic conditions promote ER stress in cardiomyocyte cultures, leading to activation of the kinase PERK. PERK phosphorylates the α-subunit of the translation initiation factor eIF2, leading to the preferential translation of the mRNA encoding activating transcription factor 4 (ATF4). In turn, ATF4 promotes the expression of several stress response genes, including REDD1. In our recent manuscript, we demonstrated that in cardiomyocytes exposed to hyperglycemic conditions, PERK inhibition or ATF4 knockdown are sufficient to prevent an increase in REDD1 expression (please see reference 9 for additional details of the prior studies).
- How would cells react after HG and both gsk3b/redd1 silencing?
Response: We have edited the manuscript to more clearly outline data supporting that REDD1 acts to promote NF-κBactivation through a signaling pathway that requires GSK3β (please see the working model in Fig. 6 and page 10 of the Discussion section, lines 297-300). The data support that GSK3β silencing or REDD1 deletion is sufficient to suppress the activation of NF-κB in cardiomyocytes exposed to hyperglycemic conditions. Indeed, the suppressive effect of REDD1 deletion on NF-κB activity in cardiomyocyte cultures exposed to hyperglycemic conditions was rescued by the expression of a constitutively active GSK3β variant. Moreover, activation of GSK3β in the heart of diabetic mice required REDD1. Together, the data support that GSK3β and REDD1 silencing result in similar outcomes with regards to the pro-inflammatory response of cardiomyocytes.
Methods:
- Why did you use different tools for silencing for gsk3b and redd1?
Response: The manuscript was edited to include additional rationale for the use of CRISPR/Cas9 to delete REDD1 and shRNA to knockdown GSK3β expression (please see lines 313-335). Frankly, the selection was based in practicality. We developed the stable REDD1-deficient AC16 human cardiomyocyte cell line in a prior study (reference 9). To evaluate a role for GSK3β, we made use of a new shRNA library that was recently made available through the Penn State College of Medicine TRC1 Human Library Informatics Core. Data herein support the efficacy of both methods.
- F12 media has 17mM glucose, which can be considered relatively high. Why did not you grow them at F12-5 mM rather than changing the media to DMEM for 2 passages?
Response: As noted by the reviewer’s comment, AC17 cells are conventionally cultured in medium with a relatively high glucose concentration, which is problematic for the evaluation of signaling changes that occur in response to hyperglycemic conditions. The protocol used for adapting AC16 human cardiomyocytes to “normal glucose” medium, so that they could subsequently be exposed to hyperglycemic conditions was based on prior published work (PMID: 31865425). We also used this method in our recent manuscript (please see reference 9). We have edited section 4.1 to clarify the rationale for the selected cell culture protocol (lines 321-326).
- Did animals change the body weight after stz+/- redd1 ko?
Response: Diabetic REDD1+/+ and REDD1-/- mice both exhibited a modest reduction in body weight as compared to non-diabetic mice. The revised manuscript includes a new graph for body weight (Fig. 5B).
Reviewer 2 Report
Comments and Suggestions for Authors
In continue to the view that stress response protein regulated in development and DNA damage response 1 (REDD1) is required for increased pro-inflammatory cytokine expression in the hearts of diabetic mice, current report used the knock-out mice to confirm it. Please conduct the concerns below.
1. In the introduction, role of REDD1 in promoting GSK3β-dependent NF-κB signaling seems not described in cardiomyocytes in clear.
2. REDD1 is essential for the sustained activation of NF-κB in cardiomyocytes. Is it observed in another cell with reference(s)?
3. PERK inhibition or ATF4 knockdown may prevent an increase in REDD1 expression in hyperglycemic cardiomyocytes. How to link it with clinical practice?
4. Current report provided the REDD1/GSK3β/NF-κB signaling in cardiomyocytes. How to apply it in clinical application?
5. Diabetic REDD1+/+ and REDD1-/- mice exhibited similar increases in fasted blood glucose concentrations. This is an interesting finding without details. Why?
6. REDD1 deletion protected diabetic mice from cardiac dysfunction that is another important view could be discussed in detail.
7. Limitation(s) of current report may strengthen the level.
Author Response
Reviewer 2: In continue to the view that stress response protein regulated in development and DNA damage response 1 (REDD1) is required for increased pro-inflammatory cytokine expression in the hearts of diabetic mice, current report used the knock-out mice to confirm it. Please conduct the concerns below.
- In the introduction, role of REDD1 in promoting GSK3β-dependent NF-κB signaling seems not described in cardiomyocytes in clear.
Response: As suggested, the Introduction was edited to more clearly outline the role of REDD1 and GSK3β in promoting NF-κB signaling in cardiomyocytes (lines 60-69). Please also see the signaling axis displayed graphically in Figure 6 for clarification.
- REDD1 is essential for the sustained activation of NF-κB in cardiomyocytes. Is it observed in another cell with reference(s)?
Response: As suggested, the manuscript was edited to highlight prior work demonstrating a role for REDD1 in sustained activation of NF-κB in retinal cells. Also please see references 10 and 11.
- PERK inhibition or ATF4 knockdown may prevent an increase in REDD1 expression in hyperglycemic cardiomyocytes. How to link it with clinical practice?
Response: Indeed, we previously demonstrated that PERK inhibition or ATF4 knockdown was sufficient to prevent increased REDD1 expression in cardiomyocytes exposed to hyperglycemic conditions. As suggested, we have edited the manuscript’s Discussion to more clearly link the findings with clinical practice (see line 300-303). We thank the editors for the suggestion to improve the clinical relevance of the manuscript.
- Current report provided the REDD1/GSK3β/NF-κB signaling in cardiomyocytes. How to apply it in clinical application?
Response: Please see the response to point 3 above.
- Diabetic REDD1+/+ and REDD1-/- mice exhibited similar increases in fasted blood glucose concentrations. This is an interesting finding without details. Why?
Response: The observation of a similar increase in fasted blood glucose levels in streptozotocin-induced diabetic REDD1+/+and REDD1-/- mice is consistent with prior works. Streptozotocin acts by destroying β cells of the pancreas, which are responsible for insulin secretion. Thus, mice become hypoinsulinemic and exhibit a secondary increase in blood glucose concentrations after streptozotocin administration. The finding of similar increases in fasted blood glucose in REDD1+/+ and REDD1-/- mice supports that REDD1 is not necessary for the toxic effects of streptozotocin. Importantly, the data here support that REDD1 deletion prevents the cardiac defects that are caused by elevated blood glucose concentrations. We have edited the manuscript to more clearly outline the diabetes model used herein.
- REDD1 deletion protected diabetic mice from cardiac dysfunction that is another important view could be discussed in detail.
Response: Please see the response to point 3 above.
- Limitation(s) of current report may strengthen the level.
Response: The revised manuscript now highlights key limitations of these studies. Please see the Discussion line 300-308.
Round 2
Reviewer 1 Report
Comments and Suggestions for Authors
ok, thanks for the answer